# Heterodimensional Kondo superlattices with strong anisotropy

Qi Feng[1], Junxi Duan [1]✉, Ping Wang[1], Wei Jiang [1]✉, Huimin Peng[1], Jinrui Zhong[1], Jin Cao[1], Yuqing Hu[1], Qiuli Li[1], Qinsheng Wang[1], Jiadong Zhou[1]✉ & Yugui Yao[1]✉

Localized magnetic moments in non-magnetic materials, by interacting with the itinerary electrons, can profoundly change the metallic properties, developing various correlated phenomena such as the Kondo effect, heavy fermion, and unconventional superconductivity. In most Kondo systems, the localized moments are introduced through magnetic impurities. However, the intrinsic magnetic properties of materials can also be modulated by the dimensionality. Here, we report the observation of Kondo effect in a heterodimensional superlattice $VS_2$-VS, in which arrays of the one-dimensional (1D) VS chains are encapsulated by two-dimensional $VS_2$ layers. In such a heterodimensional Kondo superlattice, we observe the typical Kondo effect but with intriguing anisotropic field dependence. This unique anisotropy is determined to originate from the magnetic anisotropy which has the root in the unique 1D chains in the structure, as corroborated by the first-principles calculation. Our results open up a novel avenue of studying exotic correlated physics in heterodimensional materials.

The electrical resistance of a non-magnetic metal usually decreases as the temperature is lowered because of the weakening of the lattice vibrations, and tends to saturate to a finite value determined by the number of defects[1,2]. However, this behavior changes dramatically when magnetic atoms are added. Rather than saturating, the resistance increases as the temperature is lowered below certain value[3]. The understanding of this famous Kondo effect reveals the importance of the interactions between the localized magnetic moment and the itinerant conduction electrons[4–6]. Such a many-body interaction develops a variety of intriguing electronic properties in materials, for instance the heavy fermion and unconventional superconductivity, which have always been the hotspots in condensed matter physics[7–11].

In most Kondo systems, the localized magnetic moments are introduced through magnetic impurities and defects[12–18]. However, the intrinsic magnetic properties of materials can also be effectively modulated by the dimensionality[19,20]. The recent report of the synthesis of the heterodimensional superlattice opens a new pathway to construct Kondo systems[21]. Different from the conventional superlattice consisting of cells with the same dimension, the heterodimensional superlattice composes components with different dimensions. It has already been observed to generate a variety of intriguing phenomena, such as exotic in-plane anomalous Hall effect[21], ultrafast charge transfer[22], and excellent microwave absorption[23]. Such a unique crystalline symmetry provides a new degree of freedom, the dimensionality, to manipulate the magnetic and charge properties in a heterodimensional superlattice, which can be called a heterodimensional Kondo superlattice (HKSL). Comparing to intercalation and doping, the localized magnetic moments and the Kondo effect in HKSLs are more controllable. Moreover, the HKSL hosts the potential to show heavy fermionic behavior. Both advantages make HKSL an ideal platform to explore exotic correlation physics.

The heterodimensional superlattices formed by transition metal dichalcogenides (TMDs) which are known to host rich magnetic properties, is a promising candidate to explore the HKSL. Here, by using $VS_2$-VS heterodimensional superlattice as one representative system, we report the observation of the Kondo effect in the HKSL. In $VS_2$-VS, there

[1]Key Laboratory of Advanced Optoelectronic Quantum Architecture and Measurement (MOE), School of Physics, Beijing Institute of Technology, Beijing, China. ✉e-mail: junxi.duan@bit.edu.cn; wjiang@bit.edu.cn; jdzhou@bit.edu.cn; ygyao@bit.edu.cn

are arrays of one-dimensional (1D) VS chains between VS$_2$ layers, as shown in Fig. 1a, with distinct local chemical environments, in which the vanadium atoms acquire the essential localized moments while the two-dimensional (2D) VS$_2$ layers host the in-plane itinerant conduction carriers. Evidenced by the logarithmic dependence of the resistance on the temperature as well as the negative magnetoresistance (NMR) under magnetic field along the three coordinate axes, the VS$_2$-VS shows the typical Kondo effect[24–26]. Interestingly, the field suppression of the low-temperature resistance upturn and the NMR are always the strongest when the magnetic field is applied along the direction of the 1D VS chains, regardless of the direction of the injected current. Such intriguing anisotropic behaviors can be attributed to the magnetic anisotropy rooted in the unique 1D chains in the heterodimensional superlattice, as corroborated by the first-principles calculation.

## Results and Discussion
### Kondo effect

We first present the data from a standard Hall bar device N1, as shown in Fig. 1b and the inset of Fig. 1c. The current is along the $y$-axis, parallel to the direction of the 1D VS chains. In Fig. 1c, the temperature dependence of the resistance reveals the metallic nature of the sample. The residual-resistance ratio is RRR = $\frac{R_{300K}}{R_{6.56K}}$ = 7.65, suggesting the presence of a trace amount of disorders. Figure 1d shows the field dependence of the Hall resistance $R_{xy}$ measured at 2 K for fields applied along the three coordinate axes as defined in Fig. 1c. Beyond the conventional out-of-plane (B//z) Hall effect, it also hosts a marked linear in-plane Hall response when the magnetic field is applied along the $x$-axis, known as a unique feature of the VS$_2$-VS heterodimensional superlattice[21,27]. Due to the existence of the mirror symmetry $M_y$, the $R_{xy}$ vanishes when the magnetic field is parallel to the $y$-axis, indicating an accurate alignment of the magnetic field to the sample.

To analyze the low-temperature transport properties of VS$_2$-VS, we plot an enlarged view of the low-temperature resistance against the temperature in a semilogarithmic way (Fig. 2a), which shows a clear upturn feature with decreasing temperature. The resistance minimum appears around 6.56 K ($T_m$) and the upturn is well captured by a linear fit to ln $T$. This logarithmic increase of the resistance with decreasing temperature is a characteristic feature of the Kondo effect, which can be suppressed by magnetic field. Indeed, the resistance upturn disappears when we gradually increase the out-of-plane magnetic field (B//z) to 2 T, as shown in Fig. 2a, ruling out the possible mechanism of electron-electron interaction, which is not affected by magnetic field[28]. Interestingly, the field suppression of the low-temperature resistance upturn shows strong anisotropy. As plotted in Fig. 2b, the resistance upturn is totally suppressed under 1 T when the magnetic field is applied along the $y$-axis (1D chain direction) which is lower than the critical field when it is along the $z$-axis.

In addition, as shown in Fig. 2c, strong NMR is observed under magnetic fields applied along all the three directions at 2 K, and shows no saturation up to 14 T. Similar to the field suppression of the low-temperature resistance upturn, the NMR also shows strong anisotropy. These behaviors of the NMR are totally different from the weak localization scenario[29], which should vanish under the magnetic field applied along the current direction. Therefore, both the low-temperature resistance upturn and the NMR evidence the existence of the Kondo effect. But, distinct from the nearly isotropic field dependence reported in other TMD compounds, the Kondo effect in VS$_2$-VS heterodimensional superlattice shows obvious anisotropic field dependence that it is almost identical when the magnetic field is applied along $x$- and $z$-axis but is stronger when the magnetic field is applied along the current direction ($y$-axis).

To quantitatively understand the Kondo effect and its anisotropic field dependence, we first investigate the low-temperature resistance upturn by fitting the temperature dependence of the resistance to the following formula[16]

$$R(T) = R_0 + bT^n + R_H(T) \qquad (1)$$

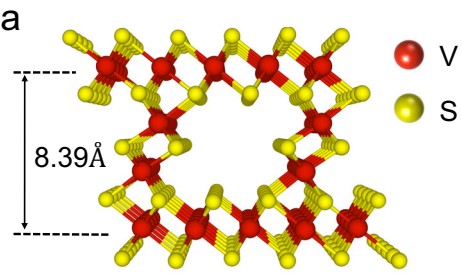

a

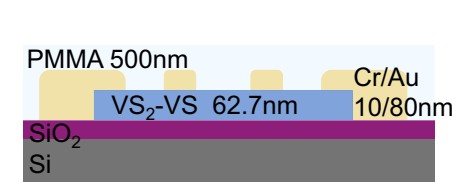

b

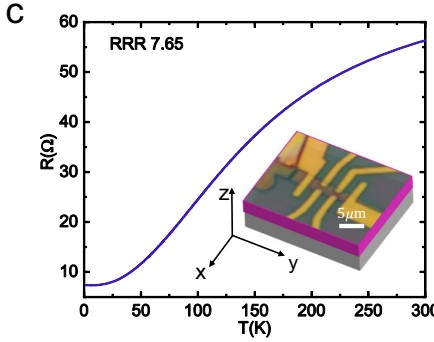

c

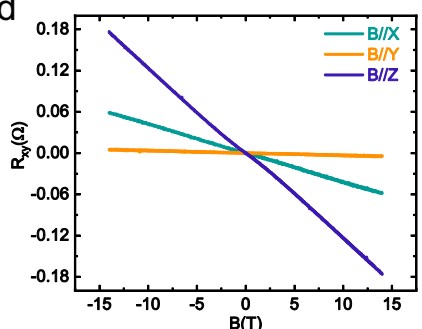

d

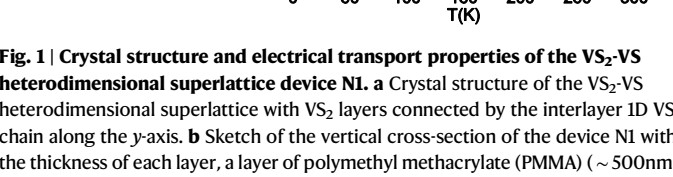

**Fig. 1 | Crystal structure and electrical transport properties of the VS$_2$-VS heterodimensional superlattice device N1. a** Crystal structure of the VS$_2$-VS heterodimensional superlattice with VS$_2$ layers connected by the interlayer 1D VS chain along the $y$-axis. **b** Sketch of the vertical cross-section of the device N1 with the thickness of each layer, a layer of polymethyl methacrylate (PMMA) ($\sim$500nm) is on the sample surface. **c** Temperature-dependent resistance for the VS$_2$-VS superlattice device N1 with its optical image shown in the inset. The current is applied along the $y$-axis. **d** Hall resistance under magnetic field along different directions at $T = 2$ K. Source data are provided as a Source Data file.

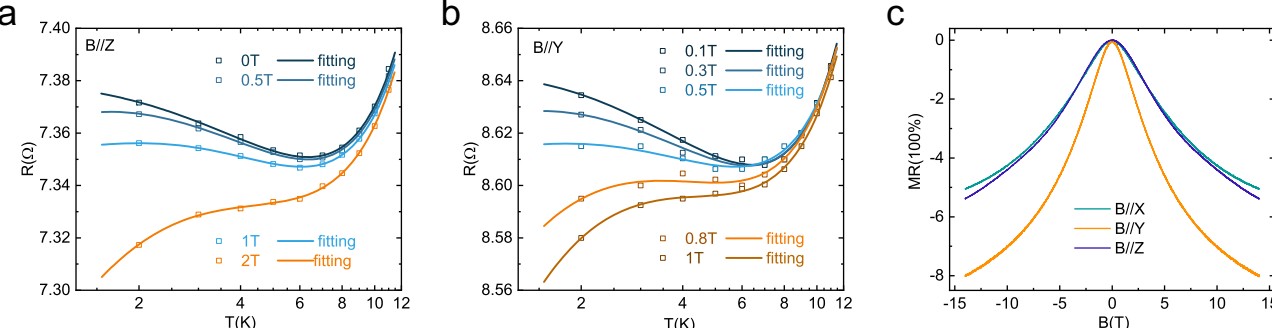

**Fig. 2 | The Kondo effect in device N1. a** and **b** Semilogarithmic plots of the temperature dependence of resistance measured under several magnetic fields applied along the z-axis (**a**) and y-axis (**b**), respectively. The solid line is the fitting of Eq. (1) with the Langevin function. **c** Negative magnetoresistance(MR)with magnetic field along different directions at 2 K. Source data are provided as a Source Data file.

where $R_0$ is the residual resistance, $bT^n$ denotes the contribution from the electron-electron and the electron-phonon interactions, and $R_H(T)$ represents the contribution from the Kondo effect described by the modified Hamann equation

$$R_H(T) = R_K \left\{ 1 - \frac{ln(T/T_K)}{[ln^2(T/T_K) + S(S+1)\pi^2]^{1/2}} \right\} \quad (2)$$

with a temperature-independent constant $R_K$, the Kondo temperature $T_K$, and the spin moment $S$ of the magnetic centers[30,31]. Here, to take the Ruderman–Kittel–Kasuya–Yosida (RKKY) interactions between the magnetic moments into consideration, the variable $T$ in the modified Hamann equation should be replaced by $T_{eff} = \sqrt{T^2 + T_w^2}$, where $k_B T_W$ represents the effective RKKY interaction strength[32]. As shown by the solid line in Fig. 2a, the experimentally measured resistance-temperature results can be well captured by the formula. From the fitting, $T_K = 7.65$K, corresponding to a Kondo energy of $J_K \sim k_B T_K = 0.66 meV$, and $S = 0.4375$ are extracted.

Essentially, Kondo resonance is a many-electron phenomenon resulting from the exchange interaction between a localized magnetic moment and the conduction electrons. When a magnetic field is applied, the splitting of the Kondo resonance appears, resulting in a suppression of the resistance upturn[33]. The magnetic field effect can be well captured by further including the Langevin function $L(x)$ to the modified Hamann expression[18]. The resistance is now described as

$$R_H(T_{eff}, B) = R_K \left\{ 1 - \frac{ln(T_{eff}/T_K)}{[ln^2(T_{eff}/T_K) + S(S+1)\pi^2]^{1/2}} \right\} \left[ 1 - L^2 \left( \frac{\mu B}{k_B T_{eff}} \right) \right] \quad (3)$$

where $\mu = g\mu_B \sqrt{S(S+1)}$ is the effective magnetic moment of the magnetic impurity, $T_K$ and $S$ are fixed with the values obtained from the zero-field fitting, and $L(x) = \coth(x) - \frac{1}{x}$ is the Langevin function. The fitted results shown in Fig. 2a, b for the magnetic field parallel to the z-axis and y-axis, respectively, which match very well with the experimental results. Not surprisingly, the Landé factor $g$ extracted from the fittings shows similar anisotropy, which is roughly 2 under magnetic field parallel to the z-axis but is around 3 when the field is along the y-axis. The anisotropic Landé factor means that the Kondo resonance peak splits at rates that are direction dependent, hinting a magnetic anisotropy in the heterodimensional superlattice.

### Anisotropic negative magnetoresistance
We now turn to the anisotropic NMR. In general, the NMR originates from the splitting of the Kondo resonance should show the same

anisotropic field dependence as the Kondo effect. However, one must be careful since there are other possible mechanisms that can contribute to an anisotropic NMR. As presented in the Hall bar device N1, the NMR is stronger when the magnetic field is applied along the current direction. It has been touted that an extra NMR induced by a magnetic field parallel to the current direction is a characteristic feature of the chiral anomaly in topological semimetals[34,35]. To testify this scenario, we prepare another device, N2, with a circular disc structure which can be utilized to investigate the dependence of the NMR on the relative direction of the magnetic field to the current. In Fig. 3a, the robust low-temperature resistance upturn exists with the minimum appears around 14.63 K ($T_m$), higher than the one in device N1 due to sample differences. We also fit the results to Eq. (1), as shown in solid line in Fig. 3a, obtaining $T_K = 9.67$K, corresponding to $J_K = 0.83$meV, and $S = 0.4236$. In the low-temperature region ($T \ll T_K$), the resistance gradually saturates with the decreasing temperature due to the disappearance of the effective localized magnetic moments by forming a nonmagnetic Kondo singlet between the localized magnetic moments and the surrounding electrons[33]. Figure 3b presents the temperature dependence of the resistance under different out-of-plane magnetic fields. Similarly, the resistance upturn is gradually suppressed with the increase of magnetic field and completely disappears under 5 T. All these features indicate that the circular disc device N2 has a similar Kondo effect as device N1. The differences in the Kondo temperature and field dependence between N1 and N2 can be ascribed to the difference in the charge carrier density (see Supplementary Note 1)[33].

Then, we measure the magnetoresistance in device N2 under magnetic fields applied in-plane but along different directions, as sketched in the inset of Fig. 3a. The current is applied along the x-axis first. As shown in Fig. 3c, we can see that the NMR is still stronger when the magnetic field is along the y-axis than the one under the magnetic field parallel to the current (x-axis). Consequently, the chiral anomaly scenario can be excluded since the NMR induced by chiral anomaly should be the largest when the magnetic field is parallel to the current. We also measure the magnetoresistance with the current in different directions ($\theta = 45°$ and 135°), as plotted in Fig. 3d. Interestingly, the NMR is always stronger when the magnetic field is applied along the y-axis, i.e. the direction of the 1D VS chain. Such a behavior unquestionably indicates that the anisotropic NMR originates from the anisotropic splitting of the Kondo resonance induced by the magnetic anisotropy. Regardless of the different direction of the injected current, the Kondo resonance splits at a higher rate when the magnetic field is applied along the 1D VS chains, resulting in a stronger NMR.

### Magnetic anisotropy
After figuring out the relation between the anisotropic NMR and the magnetic anisotropy, the NMR provides more quantitative insight into

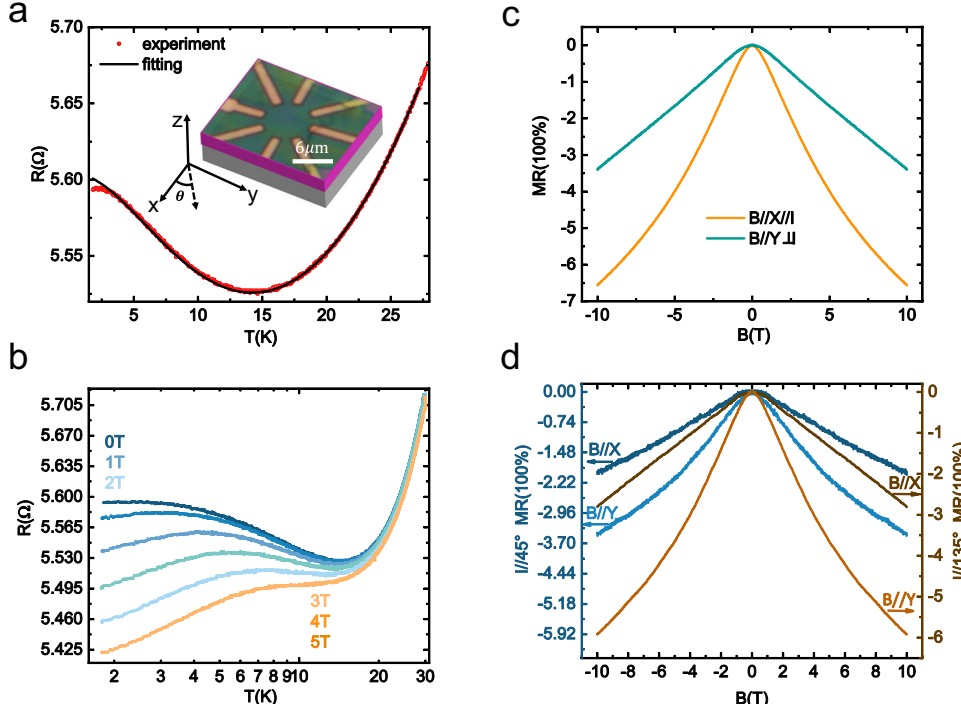

**Fig. 3 | The Kondo effect in device N2. a** The low-temperature resistance upturn measured in the circular disc device N2. The black solid line indicates the fitting curve from 1.8 to 25 K using Eq. (1). Inset: The optical image of device N2. **b** The temperature dependence of resistance measured under several magnetic fields applied along the $z$-axis. **c** Negative magnetoresistance (MR) under magnetic fields along different directions at 1.8 K when the current is applied along the $x$-axis. **d** Negative magnetoresistance under magnetic fields along different directions at 1.8 K when the current is applied along $\theta = 45°$ and 135°. Source data are provided as a Source Data file.

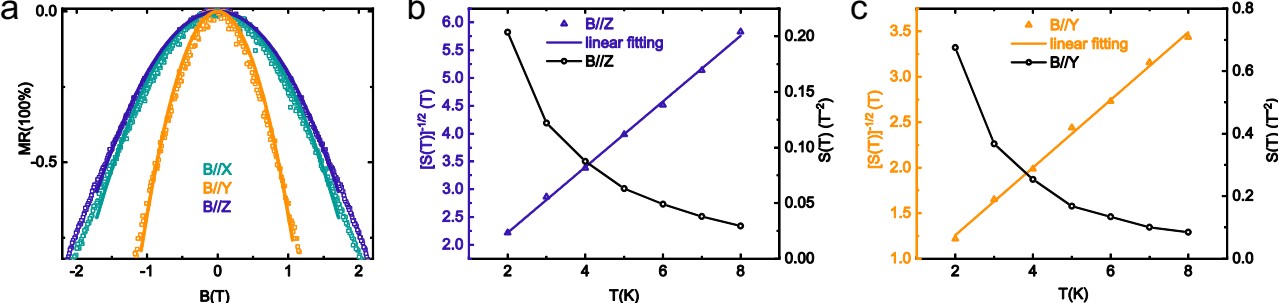

**Fig. 4 | Magnetic anisotropy. a** Magnetoresistance (MR) at 2 K with quadratic fit in solid lines. **b** and **c** Temperature dependence of $S$ and $S^{-1/2}$ from fittings of MR at low-field region when the magnetic field applied along the $z$-axis (**b**) and $y$-axis (**c**), respectively. Solid line is the linear fitting with function $S(T)^{-1/2} \propto \chi^{-1} \propto T + T_C$. Source data are provided as a Source Data file.

the anisotropic magnetization. Hence, we return to device N1. In the spin-scattering model, the NMR depends on the square of the magnetization of the localized magnetic moments. At low fields, the NMR is a parabolic function of the applied magnetic field,

$$-\frac{\triangle R}{R} = \alpha M^2 = \alpha \chi^2 H^2 = S(T) H^2 \qquad (4)$$

where the coefficient $S(T)$ is proportional to the square of the susceptibility[15,24,36,37]. In Fig. 4a, as an example, the solid lines show the fitting of the low-field NMR at 2 K under different magnetic field directions to Eq. (4). From the fittings at different temperatures, the temperature dependence of the susceptibility can be extracted. Figure 4b, c show the temperature-dependent $S(T)$ and $S(T)^{-1/2}$ when the magnetic field is applied along the $z$- and $y$-axis, respectively. It is

noted that $S(T)^{-1/2}$ depends linearly on temperature with a positive intercept. This behavior resembles the Curie-Weiss law of the low-field susceptibility of paramagnet, $\chi = \frac{C}{T + T_C}$, where $C$ is the Curie constant and $T_C$ the Curie temperature[15,24]. From the fitting, $T_C$ is positive, suggesting antiferromagnetic (AFM) coupling between the localized magnetic moments for $T < T_C$. The values of $T_C$ equal to 1.75K and 1.39K, corresponding to anisotropic AFM exchange energies ($J_{AFM} \sim k_B T_C$) of 0.15meV and 0.12meV, for the magnetic field parallel to the $z$- and $y$-axis, respectively. The lower $T_C$ for the $y$-axis indicates that the localized magnetic moments are more easily aligned by magnetic field along the 1D VS chains, consistent with the larger Landé factor.

We will discuss the origin of the Kondo effect with anisotropic field dependence and the anisotropic magnetization. We note that the Kondo effect has been observed in several V-intercalated $VX_2$ ($X =$ Se,

Te) and self-intercalated $V_xS_y$ systems[15–18], where the localized magnetic moments are introduced through interstitial V ions. However, in our $VS_2$-VS heterodimensional superlattices, there is no intentional V intercalation[21]. Meanwhile, the Kondo effect induced by interstitial V ions shows isotropic field dependence, which is different from our observations. We next examine possible origins from the $VS_2$ layer. First, we can rule out the origins from point and line defects in the $VS_2$ layer. In our $VS_2$-VS heterodimensional superlattice, the $VS_2$ layer is in 1 T phase which hosts a C3 rotation symmetry. The point defects in $VS_2$ should at least have an isotropic in-plane field dependence. For the line defects, although they are anisotropic locally, the C3 symmetry requires that the line defects along the three principal axes are equivalent. As a result, from the viewpoint of the sample, all the line defects as a whole should also show isotropic in-plane field dependence. This agrees with the results in the V-intercalated $VS_2$ that the field dependence is isotropic. Therefore, the point and line defects in the $VS_2$ layer cannot explain our observations. Second, we can rule out the possible origin from local magnetic moments in the $VS_2$ layer induced by charge transfer caused by the 1D VS chains. In general, it is possible to induce local magnetic moments if the electrons are localized by quantum confinement or correlations, such as in quantum dots or flat-band systems[38,39]. However, the $VS_2$ layer in our $VS_2$-VS heterodimensional superlattice is in 1 T phase, which is a normal metal. As a result, the electrons cannot be localized in the $VS_2$ layer to induce nonzero local magnetic moments even if there is charge transfer caused by the 1D VS chains.

Therefore, the distinct anisotropic behaviors observed in the $VS_2$-VS heterodimensional superlattice, especially that the responses are the strongest when the magnetic field is applied along the 1D VS chains, indicate the connection between the Kondo effect and the unique crystalline structure. By stacking arrays of 1D VS chains between the 2D $VS_2$ layers, localized magnetic moments with anisotropic magnetization are introduced into the structure, forming a HKSL. We can further exclude possible origin from line defects in the 1D VS chains. Their concentration is extremely low since they have not been seen by the annular dark-field scanning transmission electron microscopy in our previous study[21], which suggests that the line defect has a high formation energy in 1D VS chains.

To corroborate the unique effect of the 1D VS chains, we carry out first-principles calculations of the $VS_2$-VS heterodimensional superlattice (see Methods for calculation details). Different magnetic configurations are calculated to study the magnetic ground states and also to analyze magnetic anisotropy with different Hubbard-$U$ corrections tested. The magnetization is found to be mainly concentrated on the vanadium atoms regardless of vast variety of magnetic configurations. AFM ground state has been verified, which is consistent with previous reports[27]. The magnetization for the V atoms while choosing $U_{eff} = 4$ eV is about $1.65\mu_B$, in agreements with the experimental result, which increases slightly with the increasing of the Hubbard-$U$ values. We further calculate the magnetic anisotropy for the ground state AFM configuration, which shows a clear anisotropic magnetization with the energy of magnetization along the $y$-axis lower (about $0.5$meV/formula) than that of both $x$- and $z$-axis (see Supplementary Note 2). To further unravel the underlying origin of this intriguing feature, we analyze the orbital-resolved density of states near the Fermi level, which are mainly contributed by the $dxy$ and $dyz$ orbitals from the V atoms. The large contribution of the $y$ component orbitals can be attributed to the influence of the interlayer 1D VS chain along the $y$-axis, which yields the magnetization easy axis along the same axis. In addition, we calculate the magnetic anisotropy of the ferromagnetic configuration, which shows similar results (see Supplementary Note 2). Such a robust magnetic anisotropy against different magnetic configurations strongly demonstrates the close connection between the anisotropic magnetization and the unique structural feature of the heterodimensional superlattice induced by the interlayer 1D VS chains. When there is a magnetic field along the easy magnetic axis, the localized spins are more easily polarized, facilitating the Kondo resonance splitting, as verified in various theoretical calculations based on numerical renormalization group method[40,41]. Given that the magnetic easy axis for the $VS_2$-VS is along the VS chains, i.e. the $y$ direction, the Kondo resonance splitting is enhanced when the magnetic field is applied along the $y$ direction, resulting in the observed anisotropic field dependence of the Kondo effect.

In conclusion, we have systematically studied the Kondo effect in a HKSL candidate $VS_2$-VS. The low-temperature resistance upturn, can be suppressed under the applied magnetic field, and negative magnetoresistance are observed, which are characteristic signatures of the Kondo effect. In addition, the field suppression of the low-temperature resistance upturn and the NMR show strong anisotropy. With a circular disc device, the anisotropic NMR is determined to originate from the anisotropic splitting of the Kondo resonance induced by magnetic anisotropy rather than the chiral anomaly. From the extracted Landé factor and Curie temperature, the anisotropic magnetization caused by the unique crystalline structure of the HKSL is confirmed that the localized magnetic moments have the root in the 1D VS chains and are more easily aligned when the magnetic field is parallel to the 1D VS chains, which is supported by the theoretical calculations. Our observations shed light to the opportunity to explore the correlation physics in the intrinsic heterodimensional Kondo superlattice.

## Methods
### Device preparation
The V-based superlattice $VS_2$-VS was successfully synthesized by a molten-salt chemical vapor deposition method. The unique crystal structure has been identified by annular dark-field scanning transmission electron microscopy[21]. To investigate the novel properties of the superlattice $VS_2$-VS, Hall bar and circular disc devices were fabricated by standard electron beam lithography (EBL) technique, followed by electron-beam evaporation of 10/80 nm Cr/Au metal stacks as contact electrodes. After the device fabrication, we spin-coated a layer of polymethyl methacrylate (PMMA) (~500nm) on the sample surface to avoid degradation. The insets in Figs. 1c and 3a show the optical images of device N1 and N2. The thickness of the $VS_2$-VS flake is about 63 nm in device N1, and 102 nm in device N2. In addition, to ensure that the devices are $VS_2$-VS heterodimensional superlattices, we have measured the in-plane anomalous Hall effect, a unique transport property of such an intriguing structure, of each device.

### Measurements
The electrical transport measurements were performed using a lock-in method in a physical property measurement system (PPMS, DynaCool Quantum Design). All the measurements are done in a four-probe geometry to eliminate the electrode-sample interface effect. The contact is ohmic with a typical contact resistance of 15Ω. To eliminate the error induced by the misalignment of electrodes, symmetrized (antisymmetrized) procedures were carried out.

$$R_{xy}^{sym} = \left[ R_{xy}(\mu_0 H) - R_{xy}(-\mu_0 H) \right]/2 \quad (5)$$

$$R_{xx}^{sym} = \left[ R_{xx}(\mu_0 H) + R_{xx}(-\mu_0 H) \right]/2 \quad (6)$$

$$MR = \frac{\left[ R_{xx}^{sym}(\mu_0 H) - R_{xx}(0) \right]}{R_{xx}(0)} \times 100\% \quad (7)$$

### DFT calculations
The theoretical calculations were performed using first-principles methods based on the density functional theory. The generalized gradient approximation exchange-correlation potentials plus the

projector augmented wave method for the electron-ion interaction was used[42], as implemented in Vienna ab initio simulation package code[43]. All self-consistent calculations were performed with a plane-wave cutoff of 400 eV. The geometric optimizations were carried out without any constraint until the force on each atom is less than 0.001 eV/Å and the change of energy per cell is smaller than $10^{-6}$eV. The Brillouin zone k-point sampling was set with a $7 \times 7 \times 7$ Γ -centered Monkhorst-Pack grids. To better describe the localized 3*d* electrons of transition metal, an additional on-site Hubbard-*U* correction term was applied with the different *U* values tested. The spin-polarized calculations were performed to obtain the electronic structure of ferromagnetic (FM) and anti-ferromagnetic (AFM) configurations. The spin-orbit coupling effect was considered to estimate the Magnetocrystalline anisotropy energy (MAE).

### Reporting summary

Further information on research design is available in the Nature Portfolio Reporting Summary linked to this article.

## Data availability

The data generated during this study is provided in the main article and the Supplementary Information. The data underlying the figures are provided in the Source Data file. Source data are provided with this paper.

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

## Acknowledgements

The research was supported by the National Key R&D Program of China (Grants No. 2020YFA0308800, No. 2019YFA0308402), the National Natural Science Foundation of China (Grants No. 12350402, No. 12204037, No. 62174013, No. 92265111, No. 12074036), the Beijing

Natural Science Foundation (Grant No. Z190006), the Strategic Priority Research Program of Chinese Academy of Sciences (Grant No. XDB30000000), the funding Program of BIT (Grants No. 3180012212214, No. 3180023012204). The fabrication was supported by Micro-nano fabrication center of Beijing Institute of Technology. W. J. thanks the support from the Beijing Institute of Technology Research Fund Program for Young Scholars.

## Author contributions

J.D. and Y.Y. conceived the project. J.D. designed the experiment. Q.F. and J.D. conducted the experiment with the help from H.P., J.Zhong., Y.H., Q.L., and Q.W. J. Zhou. and P.W. grew the samples. W.J. and J.C. performed the first-principles calculations. J.D., W.J., and Q.F. wrote the manuscript with the inputs from all the authors.

## Competing interests

The authors declare no competing interests.
