## [Peer Review File · Nature Communications]

REVIEWER COMMENTS

Reviewer #1 (Remarks to the Author):

The authors report the observation of the Kondo effect in a 1D VS chain sandwiched between 2D VS₂ layers. They observed that both the temperature-dependent resistivity and negative magnetoresistance agree with the Kondo effect. Furthermore, they show that the observed Kondo effect exhibits a unique magnetic anisotropy, which they attribute to the larger Kondo resonance splitting when the magnetic field is applied along the 1D axis direction.

Overall, the results showing that the Kondo effect can be achieved through dimensionality engineering instead of using magnetic impurity is interesting and of great value to the community working on 2D heterostructure. I recommend publication in Nature Communications after the following comments are addressed:

1. The information regarding the device structure and characterization is lacking. The authors should draw a schematic showing the vertical cross-section of their device, including the thickness of each layer. They should mention how they can ensure that the VS₂ is a monolayer and the VS chain is a single layer. Also, the device size is not mentioned. They should put scale bars in the optical microscope images in Fig. 1b and 4a. They also should mention if there is any insulator encapsulation (e.g. hBN) used.
2. Is there any transport measurement on a VS₂ monolayer or bilayer device to show that the Kondo effect is only observed in a heterodimensional structure? Such base measurement is important to rule out defects contribution (either line or point defect) to the Kondo effect.
3. The authors should briefly discuss the contribution of the contact resistance and the electrode-sample interface.
4. The authors should discuss the significance of Supplementary Information Fig. S1 and S2 in relation to the claim that the Kondo resonance splitting is enhanced for the magnetic field in the y direction.
5. The following sentences are confusing:

“One should also note that, similar to the field suppression of the low-temperature resistance upturn, the NMR also shows strong anisotropy. Therefore, both the low-temperature resistance upturn and the NMR evidence the existence of the Kondo effect.”

These sentences can mislead readers that there is a relationship between Kondo effect and anisotropy. As far as I know, these two have no clear relationship.

6. For clarity, in Fig. 1a, the authors should indicate which one is the V atom and which one is the S atom.

Reviewer #2 (Remarks to the Author):

The authors reported observation of Kondo effect with strong spatial anisotropy in the VS₂-VS hybrid superlattice by transport measurement. By measuring two devices with different geometry, they confirmed the intrinsic anisotropy of the Kondo effect and linked it with the unique crystalline structure of the material. The manuscript is logically written and the experimental results are carefully analyzed. However, Kondo effect has been reported in various 2D materials. Also, the anisotropic response of the resistance and the negative magnetoresistance with external field is also not surprising given the stacked 1D structure of the VS layer. Therefore, the manuscript is not suitable for publication in high profile journals such as Nature Communications in respects of novelty and broad interest. After addressing the following issues, the manuscript could be published in more specialized journals.

1) The authors claimed that the local moments are confined in the 1D VS chains. However, related experimental evidence is lacking. While the anisotropy of Kondo resonance links the local moments to the 1D channels, they could be produced by the line defects of VS chains or reside in the VS₂ layers due to charge transfer. The authors should provide experimental evidence to corroborate their conclusion.

2) If the Kondo effect is an intrinsic property of the VS₂-VS superlattice, why do the authors expect such a big difference in Kondo temperatures in the two devices?

3) Related to the last point, is the extracted Kondo temperature reproducible in devices with similar geometries?

Reviewer #3 (Remarks to the Author):

The authors have performed directional magneto-transport measurements on heterodimensional superlattice system VS₂-VS. They have demonstrated the suppression of resistivity upturn upon application of magnetic field, as well as anisotropic negative magnetoresistance (NMR). To rule out chiral anomaly, they have also show that device having different geometry also demonstrate similar anisotropic NMR. These results build on authors' discovery of in-plane anomalous Hall effect on the same system (Nature 609, 46–51 (2022)), to emphasize upon rich many body physics present in 2D-1D superlattices. Even though this system is of significant intrigue, there are several drawbacks in the data and its presentation that needs to be addressed before the results are accepted for publication. I note then below:

1. Authors have provided Kondo temperatures of 7.7 K and 9.67 K for devices N1 and N2 respectively. There seems to be a significant difference in temperature corresponding to the resistance minima in devices N1 and N2, from Figs 2 (a) and 3 (a) respectively. Also, there seems to be different field responses for $B//z$, in N1 the resistance upturn is completely suppressed at 2T whereas for N2 it is 5T. Can the authors provide an explanation for this?

2. For the sake of clarity, can the authors provide estimated values of Kondo energy (J_K) and anisotropic AFM exchange energy (J_{AFM}) scales from their data? Would they expect the system to show heavy fermionic behavior (under e.g., tunneling spectroscopy measurement)?

3. There seems to be quite a sizeable variation from the data and its quadratic fits in Fig 4 (a), which is highest for $B//x$, is there a physical reason behind this?

Below we provide the point-by-point responses to reviewers' comments.

Reply to Reviewer #1:

0. The authors report the observation of the Kondo effect in a 1D VS chain sandwiched between 2D VS₂ layers. They observed that both the temperature-dependent resistivity and negative magnetoresistance agree with the Kondo effect. Furthermore, they show that the observed Kondo effect exhibits a unique magnetic anisotropy, which they attribute to the larger Kondo resonance splitting when the magnetic field is applied along the 1D axis direction.

Overall, the results showing that the Kondo effect can be achieved through dimensionality engineering instead of using magnetic impurity is interesting and of great value to the community working on 2D heterostructure. I recommend publication in Nature Communications after the following comments are addressed:

Response: We thank the reviewer for the careful review and recognition of our work, and we are glad that the reviewer finds our work “*interesting and of great value to the community working on 2D heterostructure*”. In the following responses, we will address all the comments in detail. We hope our responses and revisions would make the reviewer find that our revised manuscript can meet the publishing standard.

1. The information regarding the device structure and characterization is lacking. The authors should draw a schematic showing the vertical cross-section of their device, including the thickness of each layer. They should mention how they can ensure that the VS₂ is a monolayer and the VS chain is a single layer. Also, the device size is not mentioned. They should put scale bars in the optical microscope images in Fig. 1b and 4a. They also should mention if there is any insulator encapsulation (e.g. hBN) used.

Response: We thank the reviewer for pointing out these important issues. As suggested, we have added a sketch of the vertical cross-section of the device in Fig. 1b with the thickness of each layer properly labeled. The crystal structure of the samples, as shown in Fig. 1a, has been identified by annular dark-field scanning transmission electron microscopy (STEM) in our previous study [*Nature* 609, 46-51 (2022)]. Meanwhile, in-plane anomalous Hall effect (AHE) is a unique property of such a heterodimensional superlattice [*Nature* 609, 46 (2022), *PRL* 130,

166702 (2023)]. It can be used as the benchmark in transport measurements to select the right samples. We have measured the unique in-plane AHE in all the devices discussed in the manuscript to ensure that they are the heterodimensional superlattices. We have added scale bars in Fig. 1c and 3a. The samples are grown by a molten-salt CVD method on silicon wafers with an oxide layer, which are quite robust in atmosphere. However, to minimize the degradation of the devices, we spin coat a thin layer (~500 nm) of polymethyl methacrylate on the surface.

Revision: To clarify all these important details, we have revised the figures, as shown in the revised main text, and added related information in the Method section.

*In page 11, “**Device preparation.** The V-based superlattice VS_2 -VS was successfully synthesized by a molten-salt chemical vapor deposition method. The unique crystal structure has been identified by annular dark-field scanning transmission electron microscopy²¹. To investigate the novel properties of the superlattice VS_2 -VS, Hall bar and circular disc devices were fabricated by standard electron beam lithography (EBL) technique, followed by electron-beam evaporation of 10/80nm Cr/Au metal stacks as contact electrodes. After the device fabrication, we spin-coated a layer of polymethyl methacrylate (PMMA) (~500 nm) on the sample surface to avoid degradation. The insets in **Fig. 1c**, and **3a** show the optical images of device N1 and N2. The thickness of the VS_2 -VS flake is about 63 nm in device N1, and 102 nm in device N2. In addition, to ensure that the devices are VS_2 -VS heterodimensional superlattices, we have measured the in-plane anomalous Hall effect, a unique transport property of such an intriguing structure, of each device.”*

2. Is there any transport measurement on a VS_2 monolayer or bilayer device to show that the Kondo effect is only observed in a heterodimensional structure? Such base measurement is important to rule out defects contribution (either line or point defect) to the Kondo effect.

Response: We thank the reviewer for raising this insightful question. We benefit a lot from the question about possible defect contributions. The discussion of this question makes our study clearer and more rigorous.

We fully agree with the reviewer that the possibility of defect contribution can be directly

ruled out by the base measurement. However, regrettably, we have not done transport measurements on VS₂ monolayer or bilayer. The reason is that the preparation of monolayer or bilayer VS₂ is extremely difficult. To our knowledge, the reported monolayer or bilayer VS₂ studies are all done by scanning tunneling spectroscopy and angle-resolved photoemission spectroscopy in which the samples are very small [*Nat. Commun.* 12, 6837 (2021), *Npj 2D Mater. Appl.* 7, 35 (2023)]. The only reported large-scale VS₂ sample grown by CVD is in 2H phase, which is different from our 1T-phase sample [*Adv. Funct. Mater.* 30, 2000240 (2020)].

Nevertheless, the defect contributions from VS₂ layer to the observed Kondo effect with anisotropic field dependence still can be ruled out without the base measurements. In 1T-phase VS₂ monolayer, there is a C₃ rotation symmetry, which means that the linear responses to in-plane external excitations are isotropic. For the point defects, they should show isotropic in-plane field dependence. For the line defects, although they are anisotropic locally, the C₃ symmetry requires that the line defects along the three principal axes are equivalent. As a result, from the viewpoint of the sample, all the line defects as a whole should also show isotropic in-plane field dependence. What is more, according to our detailed sample analysis, the concentration of the defects is quite low that they might not be able to induce an observable Kondo behavior [*Nature* 609, 46-51 (2022)]. All these arguments agree well with reported experiments that in multilayer VS₂, the Kondo effect has only been observed in samples with V intercalation (V_{0.25}-VS₂), in which the magnetic-field dependence is isotropic [*PRB* 105, 235433 (2022)]. Here, we want to state that for real samples, there are always defects and the defects might possess nonzero local moments that potentially give rise to the Kondo effect, although no Kondo effect has been reported in VS₂ samples without intercalation. Our main point is that for those defect-induced Kondo effect, they cannot show the anisotropic field dependence as observed in our VS₂-VS sample. We think this is also what the reviewer cares about.

Revision: To make the conclusion more solid, we have added a new paragraph to discuss possible defect contributions in the Discussion section.

In pages 8 and 9, “We will discuss the origin of the Kondo effect with anisotropic field dependence and the anisotropic magnetization. We note that the Kondo effect has been observed

in several V -intercalated VX_2 ($X=Se, Te$) and self-intercalated V_xS_y systems¹⁵⁻¹⁸, where the localized magnetic moments are introduced through interstitial V ions. However, in our VS_2 - VS heterodimensional superlattices, there is no intentional V intercalation²¹. Meanwhile, the Kondo effect induced by interstitial V ions shows isotropic field dependence, which is different from our observations. We next examine possible origins from the VS_2 layer. First, we can rule out the origins from point and line defects in the VS_2 layer. In our VS_2 - VS heterodimensional superlattice, the VS_2 layer is in IT phase which hosts a $C3$ rotation symmetry. The point defects in VS_2 should at least have an isotropic in-plane field dependence. For the line defects, although they are anisotropic locally, the $C3$ symmetry requires that the line defects along the three principal axes are equivalent. As a result, from the viewpoint of the sample, all the line defects as a whole should also show isotropic in-plane field dependence. This agrees with the results in the V -intercalated VS_2 that the field dependence is isotropic. Therefore, the point and line defects in the VS_2 layer cannot explain our observations. Second, we can rule out possible origin from local magnetic moments in the VS_2 layer induced by charge transfer caused by the 1D VS chains. In general, it is possible to induce local magnetic moments if the electrons are localized by quantum confinement or correlations, such as in quantum dots or flat-band systems^{38,39}. However, the VS_2 layer in our VS_2 - VS heterodimensional superlattice is in IT phase, which is a normal metal. As a result, the electrons cannot be localized in the VS_2 layer to induce nonzero local magnetic moments even if there is charge transfer caused by the 1D VS chains. ”

3. The authors should briefly discuss the contribution of the contact resistance and the electrode-sample interface.

Response: We thank the reviewer for this suggestion. In our devices, the typical contact resistance is roughly 15Ω . Standard I-V measurements ensure that the contact is ohmic. To eliminate the contact resistance and electrode-sample-interface effect, we conducted all the measurement in a 4-probe geometry. Constant current passes the sample through the source and drain that the source-drain contact resistance does not enter the measured sample resistance. Meanwhile, the voltage drop along the sample is detected by another pair of electrodes. Since the current in the voltage detection loop is negligible, the contact resistance of the voltage probes does not enter the measured sample resistance.

Revision: As the reviewer suggested, we have added more measurement details in the Method.

In page 11, “All the measurements are done in four-probe geometry to eliminate the electrode-sample interface effect. The contact is ohmic with a typical contact resistance of 15Ω .”

4. The authors should discuss the significance of Supplementary Information Fig. S1 and S2 in relation to the claim that the Kondo resonance splitting is enhanced for the magnetic field in the y direction.

Response: We thank the reviewer for bringing up this point. Essentially, the Kondo resonance is a many-electron phenomenon resulting from the exchange interaction between a localized spin and the conduction electrons. Because of the nonzero spin or degeneracy, the localized spin become spin polarized under an external magnetic field, leading to the Kondo resonance splitting of its peak position. Therefore, when there is an easy magnetic axis, the localized spins are more easily spin polarized, facilitating the Kondo resonance splitting. There is a rather simple and intuitive physical picture that the splitting is larger in magnetically soft directions. Such a close connection between the magnetic anisotropy and the anisotropic Kondo resonance or the enhanced Kondo resonance along the easy magnetization direction is also numerically verified in various theoretical models through numerical renormalization group calculations [e.g. *Nat. Phys.* **4**, 847 (2008); *New J. Phys.* **11**, 053003 (2009)]. For the VS₂-VS heterodimensional system, the calculations have clearly shown that the magnetic easy axis is along the VS chain direction, i.e., y direction (Fig. S2 and S3 in the SI). Therefore, the Kondo resonance splitting is also enhanced when the magnetic field is along the y direction.

Revision: To better demonstrate this point, we add the aforementioned discussion in the main text with the corresponding references added.

In page 5, “Essentially, Kondo resonance is a many-electron phenomenon resulting from the exchange interaction between a localized magnetic moment and the conduction electrons.”

In page 10, “When there is a magnetic field along the easy magnetic axis, the localized spins are more easily polarized, facilitating the Kondo resonance splitting, as verified in

various theoretical calculations based on numerical renormalization group method^{40, 41}. Given that the magnetic easy axis for the VS₂-VS is along the VS chains, i.e. the y direction, the Kondo resonance splitting is enhanced when the magnetic field is applied along the y direction, resulting in the observed anisotropic field dependence of the Kondo effect.”

5. The following sentences are confusing: “One should also note that, similar to the field suppression of the low-temperature resistance upturn, the NMR also shows strong anisotropy. Therefore, both the low-temperature resistance upturn and the NMR evidence the existence of the Kondo effect.” These sentences can mislead readers that there is a relationship between Kondo effect and anisotropy. As far as I know, these two have no clear relationship.

Response: We thank the reviewer for pointing out such a crucial problem. We agree with the reviewer that these sentences are misleading. Our purpose here is to state that we have observed low-temperature resistance upturn and NMR which indicate the existence of Kondo effect, and the magnetic-field suppression of the low-temperature resistance upturn and NMR are anisotropic. In the old version, putting them together does make the reading confusing.

Revision: To make the statements clearly, we have revised related paragraphs.

In page 5, “Similar to the field suppression of the low-temperature resistance upturn, the NMR also shows strong anisotropy. These behaviors of the NMR are totally different from the weak localization scenario²⁹, which should vanish under magnetic field applied along the current direction. Therefore, both the low-temperature resistance upturn and the NMR evidence the existence of the Kondo effect.”

6. For clarity, in Fig. 1a, the authors should indicate which one is the V atom and which one is the S atom.

Response: We thank the reviewer for this helpful suggestion.

Revision: We have revised the figure accordingly. *Please see the new figure 1a in the main text.*

Reply to Reviewer #2:

0. The authors reported observation of Kondo effect with strong spatial anisotropy in the VS₂-VS hybrid superlattice by transport measurement. By measuring two devices with different geometry, they confirmed the intrinsic anisotropy of the Kondo effect and linked it with the unique crystalline structure of the material. The manuscript is logically written and the experimental results are carefully analyzed. However, Kondo effect has been reported in various 2D materials. Also, the anisotropic response of the resistance and the negative magnetoresistance with external field is also not surprising given the stacked 1D structure of the VS layer. Therefore, the manuscript is not suitable for publication in high profile journals such as Nature Communications in respects of novelty and broad interest. After addressing the following issues, the manuscript could be published in more specialized journals.

Response: We thank the reviewer for the careful review and recognition of our work, and we are glad that the reviewer thinks “The manuscript is logically written and the experimental results are carefully analyzed.”

However, we respectfully disagree with the reviewer’s criticism about the novelty and broad interest of our work. About the criticism that “*However, Kondo effect...and broad interest*”, let us first reply to the two specific points, i.e. the Kondo effect in 2D materials and the anisotropic nature of 1D structure, raised by the reviewer.

(1) The first point raised by the reviewer is that “*Kondo effect has been reported in various 2D materials.*” We agree with the reviewer that Kondo effect has indeed been observed in various 2D materials. However, most of the reported Kondo effect in 2D materials, especially those in VX₂ systems, stems from intentional dopants or interstitial atoms, and they show isotropic field-dependence. In current work, the Kondo effect is not induced by intentional doping or intercalation, and it shows anisotropic magnetic-field dependence which is closely related to the 1D chains. To our knowledge, such a unique behavior has not been reported in any other 2D materials.

(2) The second point raised is that “*the anisotropic response of the resistance and the negative magnetoresistance with external field is also not surprising given the stacked 1D structure of the VS layer.*” We fully agree with the reviewer that by stacking 1D structures, the 2D layers would possibly show anisotropic transport responses. The 1D nature should also modulate the

magnetic properties, which is exactly one of our motivations. However, such an intrinsic heterodimensional superlattice with arrays of the 1D structures encapsulated by 2D layers had not been reported before our previous work [*Nature* 609, 46 (2022)], not to mention the study of their intriguing magnetism-related transport properties. Our current work is the first to study the Kondo physics in such an intriguing heterodimensional superlattice.

We then state the innovation of our work. The Kondo effect has been a well-known and widely studied phenomenon which keeps on capturing the attention from experimentalists and theorists. In most Kondo systems, especially in the 2D Kondo systems, local magnetic moments are introduced by intentional doping or intercalation of magnetic atoms, such as iron and vanadium. However, either the doping or the intercalation is strongly sample dependent, which makes systematic understanding of the Kondo physics in these systems difficult. *In current study, we propose a heterodimensional Kondo superlattice (HKSL) in which the magnetic properties are manipulated by dimensionality.* As an example, we explore the Kondo effect in V-based transition metal dichalcogenide heterodimensional superlattice, VS₂-VS. Distinct from the reported Kondo effect in 2D materials, the most salient feature in VS₂-VS is that the field suppression of the low-temperature resistance upturn and the negative magnetoresistance are always the strongest when the magnetic field is applied along the direction of the 1D VS chain, regardless of the direction of the injected current. *Such intriguing anisotropic behaviors unquestionably indicate the connection between the Kondo effect and the unique crystalline structure, as corroborated by theoretical calculations, demonstrating the realization of a HKSL.* This intriguing structure and its unique Kondo physics have not been reported before. Comparing to doping or intercalation, by refining the growth parameters, the magnetic properties as well as the Kondo effect in the heterodimensional superlattice is more controllable. Moreover, the HKSL has the potential to show heavy fermionic behavior. Both advantages make HKSL an ideal platform to explore exotic correlation physics, which we believe will attract intense interest from material, magnetism, and strong correlation communities.

Revision: We have revised the manuscript accordingly to emphasize the novelty of our work.

In page 3, “Comparing to intercalation and doping, the localized magnetic moments and the Kondo effect in HKSLs are more controllable. Moreover, the HKSL hosts the potential to

show heavy fermionic behavior. Both advantages make HKSL an ideal platform to explore exotic correlation physics.”

1. The authors claimed that the local moments are confined in the 1D VS chains. However, related experimental evidence is lacking. While the anisotropy of Kondo resonance links the local moments to the 1D channels, they could be produced by the line defects of VS chains or reside in the VS₂ layers due to charge transfer. The authors should provide experimental evidence to corroborate their conclusion.

Response: We thank the reviewer for pointing out such a critical question. We benefit a lot from the question about possible defect contributions. The discussion of this question makes our study clearer and more rigorous.

To determine the sites of the local moments, magnetometer with atomic resolution is needed, and the spin-resolved scanning tunneling microscopy (STM) seems to be the only choice. However, the STM can only detect the surface information, especially in metallic samples, which means that it cannot locate the local moment in the 1D VS chains since they are encapsulated in 1T-phase metallic VS₂ layers. Although diamond magnetometry can sense local magnetic field, its tens-of-nanometer resolution is too coarse to locate the exact position of the local moments.

Nevertheless, we still can rule out possible origins of the observed Kondo effect with anisotropic field dependence from the line defects or charge transfer. First, the line defects of the VS chains. Although the local moments cannot be experimentally located, the line defects can be well seen through atomic-resolved annular dark-field scanning transmission electron microscopy (STEM). However, their concentration should be extremely low since such line defects had not been observed by the STEM in our previous study [*Nature* 609, 46 (2022)], which suggests the line defect has a high formation energy. Second, the local moments residing in VS₂ due to charge transfer caused by 1D VS chain. It is possible to induce local moment if electrons are localized by quantum confinement or correlations, such as in quantum dots or flat-band systems [*Nature* 391, 156 (1998), *Nat. Phys.* 10.1038/s41567-023-02360-5 (2024)]. However, in our current case, the VS₂ layer is in 1T phase that is just a normal metal. As a result, the electrons cannot be localized to induce nonzero local moments even if there is charge

transfer. Besides these two, we can also rule out possible origins from the intrinsic defects in the VS_2 layers. In previous studies, the Kondo effect in VS_2 samples can only be induced through V intercalation, which shows isotropic magnetic-field dependence. Therefore, the observed Kondo effect with anisotropic field dependence cannot be explained by defects and charge transfer, but only by the unique heterodimensional crystalline structure.

Revision: To make the conclusion more solid, we have added a new paragraph to discuss possible defect contributions in the Discussion section.

In pages 8 and 9, “We will discuss the origin of the Kondo effect with anisotropic field dependence and the anisotropic magnetization. We note that the Kondo effect has been observed in several V-intercalated VX_2 ($X=Se, Te$) and self-intercalated V_xS_y systems¹⁵⁻¹⁸, where the localized magnetic moments are introduced through interstitial V ions. However, in our VS_2 - VS heterodimensional superlattices, there is no intentional V intercalation²¹. Meanwhile, the Kondo effect induced by interstitial V ions shows isotropic field dependence, which is different from our observations. We next examine possible origins from the VS_2 layer. First, we can rule out the origins from point and line defects in the VS_2 layer. In our VS_2 - VS heterodimensional superlattice, the VS_2 layer is in IT phase which hosts a C_3 rotation symmetry. The point defects in VS_2 should at least have an isotropic in-plane field dependence. For the line defects, although they are anisotropic locally, the C_3 symmetry requires that the line defects along the three principal axes are equivalent. As a result, from the viewpoint of the sample, all the line defects as a whole should also show isotropic in-plane field dependence. This agrees with the results in the V-intercalated VS_2 that the field dependence is isotropic. Therefore, the point and line defects in the VS_2 layer cannot explain our observations. Second, we can rule out possible origin from local magnetic moments in the VS_2 layer induced by charge transfer caused by the 1D VS chains. In general, it is possible to induce local magnetic moments if the electrons are localized by quantum confinement or correlations, such as in quantum dots or flat-band systems^{38,39}. However, the VS_2 layer in our VS_2 - VS heterodimensional superlattice is in IT phase, which is a normal metal. As a result, the electrons cannot be localized in the VS_2 layer to induce nonzero local magnetic moments even if there is charge transfer caused by the 1D VS chains.”

In page 9, “We can further exclude possible origin from line defects in the 1D VS chains.

Their concentration is extremely low since they have not been seen by the annular dark-field scanning transmission electron microscopy in our previous study²¹, which suggests that the line defect has a high formation energy in 1D VS chains.”

2. If the Kondo effect is an intrinsic property of the VS₂-VS superlattice, why do the authors expect such a big difference in Kondo temperatures in the two devices?

Response: We thank the reviewer for the question. The Kondo effect in our VS₂-VS superlattice sample is intrinsic. The difference in Kondo temperature stems from the difference in the charge carrier densities in the two devices. According to theory, the Kondo temperature is $T_K \sim D_0 \sqrt{2J\rho} e^{-\frac{1}{2J\rho}}$, where J denotes the coupling between the itinerant electrons and the local moment, ρ the density of states of the electrons at the Fermi level [Hewson AC. *The Kondo Problem to Heavy Fermions*. Cambridge University Press (1993)]. From the formula, T_K becomes larger when ρ is increased. In our current case, device N2 has a higher carrier density, which means a larger density of states. Therefore, it has a higher Kondo temperature comparing to device N1. The difference in the charge carrier density in the two devices comes from the crystal growth process since they are from different batches, which is commonly seen in chemical vapor deposition method.

Revision: To clarify the difference of the Kondo temperature between the two devices, we add a new section “Difference of the Kondo effect in different devices” in the Supplementary.

In page 6, “The differences in the Kondo temperature and field dependence between N1 and N2 can be ascribed to the difference in the charge carrier density (see Supplementary).”

*In page 2 in SI, “1. **Difference of the Kondo effect in different devices***

The Kondo effect in our VS₂-VS superlattice sample is intrinsic. However, the two devices shown in the main text have different Kondo temperatures, and different field dependence. The differences can be ascribed to the difference of charge carrier density in the two devices.

According to theory, the Kondo temperature is $T_K \sim D_0 \sqrt{2J\rho} e^{-\frac{1}{2J\rho}}$, where J denotes the coupling between the itinerant electrons and the local moments, ρ the density of states of the electrons at the Fermi level¹. From the formula, T_K becomes larger when ρ is increased. In

our current case, device N2 has a higher carrier density, $5 \times 10^{22} \text{ cm}^{-3}$, than N1, $8.1 \times 10^{21} \text{ cm}^{-3}$, which means a larger density of states. Therefore, it has a higher Kondo temperature comparing to device N1. Additionally, we also measured another Hall-bar device N3, shown in **Fig. S1a**. Its carrier density is very close to device N1 since they are from the same piece of substrate grown in the same batch. **Fig. S1b** shows a clear resistance upturn when the temperature decreases, and the solid line is the fitting results to **Eq.1**. The extracted Kondo temperature is $T_K = 7 \text{ K}$, which is close to the one in device N1.

The Kondo temperature is extracted from fitting the resistance vs. temperature curve (RT curve). It is different from the temperature corresponding to the minimum of the RT curve. The temperature corresponding to the minimum of the RT curve, T_m , is determined by the competition between the Kondo effect and the scattering effect that it is more sample dependent.

T_K is the only scale governing the physics of the Kondo effect at low temperatures. Therefore, the different field response between the two devices can also be understood from the difference of the Kondo temperature. The magnetic field will split the Kondo resonance and destroy the Kondo effect. A critical field H_c , $H_c(T = 0) \sim 0.5k_B T_K$, is defined to characterize the effect, which agrees quite well with our measurement². For $T_K \sim 7 \text{ K}$, $H_c(T = 0) \sim 2.59 \text{ T}$. At finite temperature, $H_c \approx H_c(T = 0)$ if $T < 0.25 T_K$ but becomes smaller at higher temperature. The formula also shows that the Kondo effect in samples with higher Kondo temperatures can survive under higher magnetic fields. In our case, device N1 has a lower Kondo temperature than device N2. Therefore, the Kondo effect in N1 can be destroyed by a lower field comparing to N2.”

3. Related to the last point, is the extracted Kondo temperature reproducible in devices with similar geometries?

Response: We thank the reviewer for the question. Yes, we had measured another Hall-bar device N3. The extracted Kondo temperature is $T_K \sim 7 \text{ K}$, which is close to the one in device N1. Regrettably, the sample was damaged by electric shock during sample change before we measured the carrier density. However, its carrier density should be very close to device N1 since they are from the same piece of substrate grown in the same batch. On the contrary, device N2 is from another batch in which the growth parameters have been refined to obtain samples

with larger scale. This is also the reason that device N2 has a different carrier density from N1.

Revision: We have added the results from N3 in the new section as mentioned in question #2 in Supplementary. *The data from N3 is shown in Fig. S1 in the SI.*

Reply to Reviewer #3;

0. The authors have performed directional magneto-transport measurements on heterodimensional superlattice system VS₂-VS. They have demonstrated the suppression of resistivity upturn upon application of magnetic field, as well as anisotropic negative magnetoresistance (NMR). To rule out chiral anomaly, they have also show that device having different geometry also demonstrate similar anisotropic NMR. These results build on authors' discovery of in-plane anomalous Hall effect on the same system (Nature 609, 46–51 (2022)), to emphasize upon rich many body physics present in 2D-1D superlattices. Even though this system is of significant intrigue, there are several drawbacks in the data and its presentation that needs to be addressed before the results are accepted for publication.

Response: We thank the reviewer for the careful review and recognition of our work, and we are glad that the reviewer finds “*this system is of significant intrigue*”. In the following responses, we will address all the comments in detail. We hope our responses and revisions would make the reviewer find that our revised manuscript can meet the publishing standard.

1. Authors have provided Kondo temperatures of 7.7 K and 9.67 K for devices N1 and N2 respectively. There seems to be a significant difference in temperature corresponding to the resistance minima in devices N1 and N2, from Figs 2 (a) and 3 (a) respectively. Also, there seems to be different field responses for B//z, in N1 the resistance upturn is completely suppressed at 2T whereas for N2 it is 5T. Can the authors provide an explanation for this?

Response: We thank the reviewer for pointing out this interesting question. The differences in the Kondo temperature and the field dependence between the two devices stems from the difference in the charge carrier density. According to theory, the Kondo temperature is $T_K \sim D_0 \sqrt{2J\rho} e^{-\frac{1}{2J\rho}}$, where J denotes the coupling between the itinerant electrons and the local moment, ρ the density of states of the electrons at the Fermi level [Hewson AC. *The Kondo Problem to Heavy Fermions*. Cambridge University Press (1993).]. From the formula, T_K becomes larger when ρ is increased. In our current case, device N2 has a higher carrier density, which means a larger density of states. Therefore, it has a higher Kondo temperature comparing

to device N1. In addition, the Kondo temperature is extracted from fitting the resistance vs. temperature curve (RT curve). It is different from the temperature corresponding to the minimum of the RT curve, T_m , which is determined by the competition between the Kondo effect and the scattering effect and is more sample dependent.

T_K is the only scale governing the physics of the Kondo effect at low temperatures. Therefore, the different field responses between the two devices can also be understood from the difference of the Kondo temperature. The magnetic field will split the Kondo resonance and destroy the Kondo effect. A critical field H_c is defined to characterize the effect. According to theory, $H_c(T = 0) \sim 0.5k_B T_K$, which agrees quite well with our measurements that for $T_K \sim 7$ K, $H_c(T = 0) \sim 2.59$ T [PRL 85, 1504 (2000)]. At finite temperature, $H_c \approx H_c(T = 0)$ if $T < 0.25 T_K$ but becomes smaller at higher temperature. The formula also shows that the Kondo effect in samples with higher Kondo temperatures can survive under higher magnetic field. In our case, device N1 has a lower Kondo temperature than device N2. Therefore, the Kondo effect in N1 can be destroyed by a lower field comparing to N2.

Revision: To clarify the difference of the Kondo temperature and the critical field between the two devices, we add a new section “Difference of the Kondo effect in different devices” in the Supplementary.

In page 6, “The differences in the Kondo temperature and field dependence between N1 and N2 can be ascribed to the difference in the charge carrier density (see Supplementary).”

In page 2 in SI, “1. Difference of the Kondo effect in different devices

The Kondo effect in our VS₂-VS superlattice sample is intrinsic. However, the two devices shown in the main text have different Kondo temperatures, and different field dependence. The differences can be ascribed to the difference of charge carrier density in the two devices.

According to theory, the Kondo temperature is $T_K \sim D_0 \sqrt{2J\rho e^{-\frac{1}{2J\rho}}}$, where J denotes the coupling between the itinerant electrons and the local moments, ρ the density of states of the electrons at the Fermi level¹. From the formula, T_K becomes larger when ρ is increased. In our current case, device N2 has a higher carrier density, $5 \times 10^{22} \text{ cm}^{-3}$, than N1, $8.1 \times 10^{21} \text{ cm}^{-3}$, which means a larger density of states. Therefore, it has a higher Kondo

temperature comparing to device N1. Additionally, we also measured another Hall-bar device N3, shown in **Fig. S1a**. Its carrier density is very close to device N1 since they are from the same piece of substrate grown in the same batch. **Fig. S1b** shows a clear resistance upturn when the temperature decreases, and the solid line is the fitting results to **Eq.1**. The extracted Kondo temperature is $T_K = 7$ K, which is close to the one in device N1.

The Kondo temperature is extracted from fitting the resistance vs. temperature curve (RT curve). It is different from the temperature corresponding to the minimum of the RT curve. The temperature corresponding to the minimum of the RT curve, T_m , is determined by the competition between the Kondo effect and the scattering effect that it is more sample dependent. T_K is the only scale governing the physics of the Kondo effect at low temperatures. Therefore, the different field response between the two devices can also be understood from the difference of the Kondo temperature. The magnetic field will split the Kondo resonance and destroy the Kondo effect. A critical field H_c , $H_c(T = 0) \sim 0.5k_B T_K$, is defined to characterize the effect, which agrees quite well with our measurement². For $T_K \sim 7$ K, $H_c(T = 0) \sim 2.59$ T. At finite temperature, $H_c \approx H_c(T = 0)$ if $T < 0.25 T_K$ but becomes smaller at higher temperature. The formula also shows that the Kondo effect in samples with higher Kondo temperatures can survive under higher magnetic fields. In our case, device N1 has a lower Kondo temperature than device N2. Therefore, the Kondo effect in N1 can be destroyed by a lower field comparing to N2.”

2. For the sake of clarity, can the authors provide estimated values of Kondo energy (J_K) and anisotropic AFM exchange energy (J_{AFM}) scales from their data? Would they expect the system to show heavy fermionic behavior (under e.g., tunneling spectroscopy measurement)?

Response: We thank the reviewer for the suggestions. We estimate the value of the Kondo energy and anisotropic AFM exchange energy from the Kondo temperature and the Curie temperature. The Kondo energy is $J_K \sim k_B T_K = 0.66$ meV for $T_K = 7.65$ K in device N1. The anisotropic AFM exchange energy, $J_{AFM} \sim k_B T_C$, is 0.12 meV for magnetic field along the 1D chain and 0.15 meV for magnetic field perpendicular to the 1D chain in device N1. For the possibility to show heavy fermionic behavior in the system, it is a very interesting question. We are glad that we have the same perspective as the reviewer and we hope the answer

is yes. As mentioned in the response to the first question, the charge carrier density has a strong effect on the Kondo effect as well as the magnetic properties. The heavy fermionic behavior might exist at some appropriate range of charge carrier density. We are currently managing to tune the charge carrier density through ionic gating in the transport devices. The tunneling spectroscopy measurement is definitely a good choice. We need to further refine our sample-growth parameters to tune the carrier density of the samples since ionic gating cannot be implemented in STM measurements.

Revision: We have added the energy scales in the main text.

In page 5, “From the fitting, $T_K = 7.65$ K, corresponding to a Kondo energy of $J_K \sim k_B T_K = 0.66$ meV, and $S = 0.4375$ are extracted.”

*In page 6, “We also fit the results to Eq. 1, as shown in solid line in **Fig. 3a**, obtaining $T_K = 9.67$ K, corresponding to $J_K = 0.83$ meV, and $S = 0.4236$.”*

In page 8, “From the fitting, T_C is positive, suggesting antiferromagnetic (AFM) coupling between the localized magnetic moments for $T < T_C$. The values of T_C equal to 1.75 K and 1.39 K, corresponding to anisotropic AFM exchange energies ($J_{AFM} \sim k_B T_C$) of 0.15 meV and 0.12 meV, for the magnetic field parallel to the z- and y-axis, respectively.”

3. There seems to be quite a sizeable variation from the data and its quadratic fits in Fig 4 (a), which is highest for B//x, is there a physical reason behind this?

Response: We thank the reviewer for this question. For NMR stemming from the Kondo effect, it shows the quadratic dependence on the magnetic field only in the near-zero-field region but will deviate significant and tend to be saturated at high-field region. In our old version, we simply chose the fitting regions by fixing the MR for magnetic field along different axes. However, the corresponding field regions are too large that the NMR starts showing deviation, as pointed out by the reviewer. In the revised version, we reduce the size of the fitting region to have a better quadratic fitting. We should emphasize that the new fitting results in slight change in the extracted Curie temperatures but does not alter the conclusion that the Curie temperature is positive and it is smaller along the y-axis.

Revision: We have revised Fig. 4 with a better choice of the fitting range. The extracted Curie temperatures in the main text are revised accordingly. *The new figure 4 please see the main text.*

In page 8, “From the fitting, T_C is positive, suggesting antiferromagnetic (AFM) coupling between the localized magnetic moments for $T < T_C$. The values of T_C equal to 1.75 K and 1.39 K, corresponding to anisotropic AFM exchange energies ($J_{AFM} \sim k_B T_C$) of 0.15 meV and 0.12 meV, for the magnetic field parallel to the z- and y-axis, respectively.”

List of changes:

(In the revised main text, we highlight the revisions in yellow.)

1.[Main Text, Page 3]

We have added some sentences to highlight the advantages of the HKSL. “*Comparing to...physics*”.

2.[Main Text, Page 5]

We have revised the sentences “*Similar to...Kondo effect*” to make the statements clear.

3.[Main Text, Page 5]

We have added the Kondo energy of device N1. “*From the fitting...are extracted*”

4.[Main Text, Page 6]

We have added a sentence to introduce the Kondo resonance. “*Essentially,...conduction electrons*”

5.[Main Text, Page 6]

We have added the Kondo energy of device N2.

6.[Main Text, Page 7]

We have added a sentence to discuss the differences between the device N1 and N2. “*The differences...(see Supplementary)*”

7.[Main Text, Page 8]

We have added the anisotropic AFM exchange energy of device N1 and revised the extracted Curie temperatures. “*From the fitting,...0.12 meV*”

8.[Main Text, Page 8 and Page 9]

We have added a new paragraph and the corresponding references to discuss possible defect contributions. “*We will discuss...VS chains*”

9.[Main Text, Page 9]

We have added sentences to discuss possible origin from line defects in the 1D VS chains. “*We can further ...VS chains*”

10.[Main Text, Page 10]

We have added sentences and the corresponding references to illustrate enhancement of the Kondo resonance splitting under magnetic field in the y direction. “*When there is ...Kondo*”

effect”

11.[Main Text, Page 11]

We have revised the “*Device preparation*” in Method to clarify important details of the device.

12.[Main Text, Page 11]

We have added sentences about the measurement details. “*All the measurements ...15Ω*”

13.[Main Text, Figure 1a]

We have revised the crystal structure of the samples to show the V and S atoms more explicit.

14.[Main Text, Figure 1b]

We have added a sketch of the vertical cross-section of device N1 with the thickness of each layer.

15.[Main Text, Figure 1c]

We have added the scale bar in the optical image of device N1.

16.[Main Text, Figure 3a]

We have added the scale bar in the optical image of device N2.

17.[Main Text, Figure 4]

We have revised the figure with a better choice of the fitting range. The extracted Curie temperatures has been revised in the main text accordingly. (also see change 7)

18.[SI, Figure S1 and Section 1]

We have added a new section “*Difference of the Kondo effect in different devices*”, including results from a new device N3, in Supplementary Information.

19.[SI, Section 2]

We have added some sentences to present Fig. S2 and S3.

20. We have also revised several typos and grammar errors to improve the English of our manuscript.

REVIEWERS' COMMENTS

Reviewer #1 (Remarks to the Author):

The authors have addressed all of my concerns. The current manuscript is suitable for publication in Nature Communications.

Reviewer #2 (Remarks to the Author):

In the revised manuscript, the authors have addressed all my questions and concerns. Importantly, they have added elaborate discussion about the novelty of the present work. Now I believe it can be accepted.

Reviewer #3 (Remarks to the Author):

The authors have satisfactorily answered all the questions raised. The paper can now be accepted for publication.